# Prolonged Intrinsic Neural Timescales Dissociate from Phase Coherence in Schizophrenia

**DOI:** 10.3390/brainsci13040695

**Published:** 2023-04-21

**Authors:** Stephan Lechner, Georg Northoff

**Affiliations:** 1The Royal’s Institute of Mental Health Research, Brain and Mind Research Institute, University of Ottawa, Ottawa, ON K1Z 7K4, Canada; 2Research Group Neuroinformatics, Faculty of Computer Science, University of Vienna, 1010 Vienna, Austria; 3Vienna Doctoral School Cognition, Behavior and Neuroscience, University of Vienna, 1030 Vienna, Austria; 4Centre for Neural Dynamics, Faculty of Medicine, University of Ottawa, Roger Guindon Hall 451 Smyth Road, Ottawa, ON K1H 8M5, Canada; 5Mental Health Centre, School of Medicine, Zhejiang University, Hangzhou 310013, China; 6Centre for Cognition and Brain Disorders, Hangzhou Normal University, Hangzhou 310013, China

**Keywords:** schizophrenia, EEG, intrinsic neural timescales, intertrial phase coherence, temporal disturbance

## Abstract

Input processing in the brain is mediated by phase synchronization and intrinsic neural timescales, both of which have been implicated in schizophrenia. Their relationship remains unclear, though. Recruiting a schizophrenia EEG sample from the B-SNIP consortium dataset (*n* = 134, 70 schizophrenia patients, 64 controls), we investigate phase synchronization, as measured by intertrial phase coherence (ITPC), and intrinsic neural timescales, as measured by the autocorrelation window (ACW) during both the rest and oddball-task states. The main goal of our paper was to investigate whether reported shifts from shorter to longer timescales are related to decreased ITPC. Our findings show (i) decreases in both theta and alpha ITPC in response to both standard and deviant tones; and (iii) a negative correlation of ITPC and ACW in healthy subjects while such correlation is no longer present in SCZ participants. Together, we demonstrate evidence of abnormally long intrinsic neural timescales (ACW) in resting-state EEG of schizophrenia as well as their dissociation from phase synchronization (ITPC). Our data suggest that, during input processing, the resting state’s abnormally long intrinsic neural timescales tilt the balance of temporal segregation and integration towards the latter. That results in temporal imprecision with decreased phase synchronization in response to inputs. Our findings provide further evidence for a basic temporal disturbance in schizophrenia on the different timescales (longer ACW and shorter ITPC), which, in the future, might be able to explain common symptoms related to the temporal experience in schizophrenia, for example temporal fragmentation.

## 1. Introduction

Schizophrenia (SCZ) is a psychiatric disorder that is characterized by positive (e.g., hallucinations and delusions) and negative symptoms (e.g., cognitive deficits in attention and working memory). Research on SCZ has been conducted on a plethora of explanatory levels, with mechanisms spanning molecular, cellular, circuit and whole-brain system levels. On the system level, electroencephalography (EEG) is a non-invasive neuroimaging technique that measures electrical activity originating in the brain. Obtained signals can be investigated using averaging techniques such as event-related potentials and time–frequency analysis. Psychiatric research uses EEG to explore electrophysiological mechanisms of disorders and develop biomarkers that can distinguish between them.

Schizophrenia patients show deficits with input processing [1,2]. One possible source of these deficits may stem from temporal imprecision on the neuronal level, which leads to problems with the temporal segregation of incoming information [3,4]. Temporal segregation requires temporal precision as the single input may be segregated from others through their distinct time points. One neuronal measure indexing temporal precision is phase coherence. Various EEG studies show that schizophrenia subjects are not able to precisely synchronize their phases to the time point of the onset of the external stimuli; this is measured by reduced intertrial phase coherence (ITPC) in particularly in theta [4,5,6], alpha [5,7], delta [4,8] and beta [9] frequencies. ITPC is a robust marker in schizophrenia but has also been reported to a lesser extent in other psychotic disorders such as bipolar and schizoaffective disorder [10,11]. Therefore, the mechanisms underlying and driving the deficit in ITPC may provide insight into disorders related to schizophrenia.

ITPC is based on the ongoing randomly distributed phase angles which are reset by the incoming external stimulus—this results in the clustering of phase angles at the hundreds-of-millisecond range after stimulus onset when averaged over trials [12]. Hence, ITPC measures the temporal consistency in the phase angles of the signal at the same time point over trials; it can therefore serve as a proxy for temporal precision and temporal segregation (e.g., segregating the stimulus onset from other time points and their related stimuli) with respect to the temporal onset of the external input [13]. Given the relatively consistent observation of decreased ITPC in SCZ, one may suppose decreased temporal precision, that is, temporal imprecision to be a driving feature in SCZ. However, where and how such temporal imprecision is coming from remains yet unclear. 

One key temporal feature of the brain’s neural activity is its intrinsic neural timescales (INTs) [14,15,16]. INTs refer to distinct temporal windows in the brain’s spontaneous neural activity by means of which it can process inputs with a variety of different timescales, e.g., tones, melodies or themes in music, through balancing their temporal integration and segregation [15,16,17,18,19,20,21]. Recent studies using fMRI demonstrate shortened temporal windows, i.e., INTs in the resting state of schizophrenia [22,23], while one EEG study demonstrated prolonged temporal windows, i.e., INTs during task states [24,25]. Here, we want to investigate whether the EEG findings of prolonged INTs, as measured by the autocorrelation window (ACW), can be extended to the resting state [14,15,18,20,24,26]. The ACW measures the rate of decay of the autocorrelation function of the signal overlayed with a copy of itself and is therefore a measure for INTs. Prolonged INTs in the resting state would mean that the balance is shifted towards temporal integration (at the expense of temporal segregation) already in the spontaneous neural activity, resulting in higher degrees of temporal imprecision in input processing. Therefore, both INTs and ITPC are measures of input processing, but INTs assessed during rest serve as measures of precondition, while ITPC, which is measured during task, is concerned with the direct neural response to the stimulus. The main goal of our paper therefore is to investigate how the changes in INTs in SCZ are related to temporal imprecision as indexed by ITPC? 

### Aims

Our first specific aim is to corroborate existing studies that show decreases in temporal synchronization, i.e., ITPC in schizophrenia. Based on previous findings [4], we hypothesized decreased ITPC in particular in theta and alpha bands to be present in basically all EGG channels (to a lesser or stronger degree). A second specific aim is to provide evidence for the changes in ACW in schizophrenia, both during an auditory oddball task and more importantly in a resting-state recording. Building on and extending the previous EEG findings [24], we hypothesized an abnormally long ACW in SCZ in schizophrenia compared to healthy subjects. This would indicate that the balance of temporal integration versus segregation of input processing is abnormally shifted towards the former in schizophrenia. The third specific aim of our study is to relate the two measures by looking at the correlation of ACW (in both the rest and task states) with both standard and deviant ITPC. We hypothesize a correlational relationship, of ACW and ITPC in healthy subjects, while this relationship may no longer be present in schizophrenia as abnormally prolonged ACW may make impossible the exact timing required for phase coherence as manifest in the decreased ITPC.

## 2. Materials and Methods

### 2.1. Participants

A total of 134 age-matched participants were included in the current study. We analyzed 70 schizophrenia patients (22 female and 48 male, age = 36.56 ± 1.31 years) and 64 healthy control subjects (HC; 37 female and 27 male; t = 0.596, *p* = 0.552). An overview of the sample and its demographic makeup, as well as information on diagnosis, can be seen in Table 1; a more in-depth discussion of recruitment can be found in Tamminga et al. 2013 [27]. The sample was selected based on three criteria: (1) EEG recordings must have been available for both the auditory oddball and the resting-state sessions, (2) recordings must have been performed using 64 electrodes and (3) participants’ resting-state recording could not be shorter than 5 min. Healthy control subjects had no history of psychiatric disorders as assessed by the Structured Clinical Interview for Diagnostic and Statistical Manual of Mental Disorders, Fourth Edition (DSM-IV) and had no first-degree relatives that had a diagnosis of any mental disorder. This study was approved by Institutional Review Boards of the respective B-SNIP consortium site and all study participants gave written informed consent prior to participation. The University of Ottawa Institute of Mental Health Research REB approved of the data sharing (REB # 2021002).

### 2.2. EEG Recordings

EEG was recorded from 64 channels (Ag/AgCl electrodes, impedance < 5 KΩ) via a Neuroscan Quick Cap (Compumedrics, El Paso, TX, USA) with a mid-forehead ground and nose reference. Data were acquired at a sampling rate of 1000 Hz, digitally amplified 1000 times and filtered using a high-pass DC filter and a low-pass filter at 2000 Hz. Data collection setups were identical in the exact specifications and were operated by specially trained personnel at all recording sites to guarantee data quality consistency across sites. Electrodes were placed on the scalp based on the 10-10 system. Resting-state EEG was recorded for 5 min per subject while participants were instructed to keep their eyes closed. For the oddball task, subjects were presented with 667 randomized auditory stimuli, 85% (567) of which were 1000 Hz sinusoidal standard tones and 15% (100) were 1400 Hz sinusoidal deviant tones. Each stimulus was presented for 50 ms with a rise and decay of 10 ms. The presentation of the tones was randomly interspersed with an inter-stimulus interval fixed to 1300 ms. Participants were instructed to respond as fast as possible at the occurrence of the deviant tone and to additionally keep count of the occurrences. For analysis of the ACW, the middle 5 min of the recording were taken out to match the length of resting-state recordings. ACW was collected for both resting and task states, ITPC was collected from the task state exclusively.

### 2.3. Preprocessing

Raw data were down-sampled to 500 Hz and then preprocessed using the Harvard Automated Preprocessing Pipeline for EEG (HAPPE) [28] in EEGLAB v2018b [29], following the procedure in Northoff et al. 2021 [24] with minor alterations. As in Northoff et al. (2021) [24], notch filtering at 60 and 120 Hz instead of HAPPE’s cleanline method was used to suppress line noise. This substitution was chosen because it has yielded better effects in suppressing line noise than the original cleanline procedure in the previous study. In contrast to Northoff et al. (2021) [24], we used the FastICA instead of the runICA function to remove artifacts, to save time and deal with instabilities in the runICA algorithm. Continuous recordings were segmented into two-second windows to perform the steps in HAPPE. To preserve the continuity of the time series needed for the calculation of ACW, no bad segments were excluded from the analysis. Instead, we used the FASTER method to repair bad segments [30]. Finally, data were re-referenced to a common average reference and data was bandpass filtered between 1.3 and 50 Hz using an FIR filter.

### 2.4. Analysis

#### 2.4.1. Intertrial Phaser Coherence

For analysis of intertrial phase coherence (ITPC), the continuous time series was segmented into epochs from 800 ms before to 1100 ms after stimulus onset. ITPC was then obtained using Morlet Wavelet convolution implemented in MNE using the time_frequency.tfr_morlet function, with the number of cycles set to half of each frequency [31]. Forty logarithmically scaled frequencies between 1.3 and 50 Hz were chosen for ITPC extraction. Based on visual inspection of values of healthy participants, ITPC values were averaged for theta and alpha frequencies (3 Hz to 8 Hz) and between 50 and 300 ms after onset (see rectangle in Figure 1A,B). ITPC was calculated for 100 standard and 100 deviant trials.

#### 2.4.2. Autocorrelation Window

The autocorrelation function was obtained for each subject and channel using the fast Fourier transform algorithm as implemented in statsmodels library for python [32]. The lag was set to the duration of the entire time series (5 min). In a next step, the autocorrelation window (ACW) was defined as the full width at half maximum of the main lobe of the autocorrelation function (Figure 2A) [18,20,26]. For both resting and task states, we used a dynamic calculation for the ACW in which we segmented the time series in 22 non-overlapping 13-s windows for which ACW values were obtained. The average of the segments’ ACW was then defined as the value based on which statistical analysis was performed. Task-state ACW was calculated from the 5 min in the middle of the recording to match the length of the resting state.

The difference between resting and task-state ACW was calculated for each electrode by subtracting task ACW from rest ACW, i.e., rest–task difference. A negative value of the rest–task difference in any given electrode, therefore, means that ACW in the task state was longer than the resting-state ACW for that electrode. Conversely, a positive value means that resting state ACW was longer than task ACW (see Figure 2C (right)).

#### 2.4.3. Statistical Analysis

All statistical analyses were performed in R v4.0.2, python 3.6. Since we had a clear hypothesis of prolonged ACW in both the rest and task states, and decreased ITPC in schizophrenia, we used one-sided *t*-tests to test for calculating group differences. For group comparisons of rest–task difference, a two-sided *t*-test was used, because this step did not follow an a priori hypothesis but was more exploratory. Pearson correlation was used to test for relationship of ACW and ITPC. All analyses were performed on central electrode Cz.

## 3. Results

### 3.1. Decreased ITPC in Schizophrenia

Intertrial phase coherence was significantly decreased in schizophrenia for standard trials (*M* = 1.13 ± 1.78) compared to healthy patients (*M* = 0.23 ± 1.27; *t*_133_ = 4.8, *p* < 0.001). The same pattern was observed for deviant tones, where schizophrenia was again decreased (*M* = 0.25 ± 0.08) compared to healthy participants (*M* = 0.32 ± 0.1; *t*_133_ = 3.805, *p* < 0.001). Notably, as can be seen in our topo plots, this seems to hold over all electrodes to a lesser or stronger degree.

### 3.2. Longer ACW during the Resting State in Schizophrenia

Group differences were present for the resting state (Figure 2). The length of ACW during the resting state was significantly longer in schizophrenic subjects (*M* = 11.87 ± 2.27 ms) when compared to healthy controls (*M* = 11.02 ± 1.78 ms; *t*_133_ = t = −2.164, *p* = 0.016; Figure 2C). While present across the scalp (see topo maps in Figure 2), this prolongation was most pronounced in frontal and central electrodes (Figure 2B). During task, duration of ACW was not significantly extended in the schizophrenia group (*M* = 1.74 ± 1.57 ms) when compared to healthy controls (*M* = 1.78 ± 1.66 ms, *t*_133_ = 0.365, *p* = 0.64; Figure 2D). Finally, rest–task differences in ACW were significantly smaller in schizophrenia (*M* = 1.13 ± 1.78 ms) than in healthy controls (*M* = 0.24 ± 1.27 ms, *t*_133_ = −3.379, *p* = 0.001). Mean values for healthy controls showed smaller values compared to those in schizophrenia, meaning that average task-state ACW was longer compared to ACW in the resting state. Conversely, schizophrenia patients did not show such task-related ACW prolongation entailing that, unlike in healthy subjects, their rest–task difference was tilted towards longer resting-state ACW. In sum, ACW is lengthened in schizophrenia during the resting state and shows reduced rest–task difference.

### 3.3. Negative Relationship between ACW and ITPC

For healthy participants, ACW at rest shows a negative correlation with ITPC for standard (R = −0.52, *p* < 0.001 Figure 3A) and deviant (R = −0.28, *p* = 0.025, Figure 3B) tones. In contrast, this was not the case for schizophrenia patients (standard: R = −0.12, *p* = 0.34; deviant: R = −0.003, *p* = 0.98, Figure 3A and Figure 3B, respectively). Interestingly, the ACW task also showed the same pattern, despite not showing significant group differences on its own. Healthy participants showed a negative correlation of ACW task with ITPC standard (R = −0.44, *p* < 0.001, Figure 3C) and deviant (R = −0.28, *p* = 0.024, Figure 3D), while this relationship was again absent for schizophrenia patients (standard: R = −0.137, *p* = 0.26; deviant: R = −0.051, *p* = 0.673; Figure 3C and Figure 3D, respectively). Rest–task difference of ACW did not show any significant relationship with neither standard nor deviant ITPC (Appendix A). In sum, ACW is related negatively to ITPC in healthy subjects whereas that is no longer the case in schizophrenia.

## 4. Discussion

We have here demonstrated deficits in temporal precision in schizophrenia along two lines. Our first main finding shows decreased intertrial phase coherence in high theta and low alpha frequency bands (Figure 1). This extends previous findings [4,5,7,8] by showing reduced phase synchronization during both deviant and standard tones. ITPC is believed to assess the brain’s capacity to temporally align to incoming stimuli, by resetting the phase of ongoing oscillations to the input, a process also referred to as entrainment [13]. This resetting has been implicated in various cognitive processes, such as speech perception, attention and the integration of cross-modal inputs [33,34,35] and has been suggested as major biomarker in SCZ [36]. The reported reduction in phase synchronization therefore suggests deficits in temporal segregation of the external tones from the ongoing internal activity as the former can no longer be temporally distinguished from the latter by phase resetting to the temporal onset of the external stimulus. This is not just the case for deviant stimuli that require the allocation of attentional resources. On the contrary, in our findings, deficient alignment to the stimulus is even more pronounced for standard tones, which could mean that unexpected information can be processed to a higher degree than ordinary, irrelevant or background information. Together, these findings suggest decreased temporal segregation of the external input resulting in higher degrees of temporal imprecision. 

The interpretation of decreased temporal segregation is supported by our second finding, the imbalance between short and long intrinsic neural timescales as measured by the autocorrelation window (Figure 2). In particular, we show that this imbalance is tilted from short INTs towards long INTs in schizophrenia. This is in line with a recent EEG study which also demonstrated abnormally prolonged ACW during a self-enfacement task [24]. Our results extend these findings to a different paradigm, that is, oddball rather than self-enfacement, and the resting state. Research at lower frequencies using fMRI has found that different gradients of INTs are present in different sensory processing hierarchies in schizophrenia [22,23]. Particularly, auditory pathways showed increasingly long timescales the higher up a region was in the hierarchy. Contrary to our findings, Wengler and colleagues found that the increase in timescale length along the auditory pathway was less pronounced in SZ patients compared to HC participants, while a moderate increase was observed in the somatosensory input stream. The reduction in INTs in auditory areas was further correlated with severity of hallucinations and delusions [23]. A similar reduction in INTs of resting-state fMRI of primary sensory areas was found in high-functioning autism patients [37]. The causes for the opposite direction for different frequency ranges (low fMRI frequencies vs. fast EEG frequencies) remains to be explored. However, given that the prolonged ACW in EEG is already present in the resting state, any input processing during task states may be affected by it. That is exactly what we observed in our third main finding, the decoupling or dissociation of ITPC from ACW. To fully understand this finding, we need to make a brief detour into the role of ACW for input processing. 

INTs concern the brain’s intrinsic propensity for temporal segmentation of the input sequences. By employing its own temporal windows, the brain can segment the continuous inputs into meaningful chunks [14,15,16]. This makes possible the temporal integration and segregation of the incoming inputs by the brain’s neural activity [18]. If the temporal windows of the neural activity are too long, several incoming inputs are lumped together through increased temporal integration while, at the same time, temporal segregation is reduced. This scenario we assume to hold in schizophrenia. The prolonged ACW increases temporal imprecision through increased temporal integration of the external inputs’ timing onset with the adjacent time points of the ongoing internal activity. That, in turn, makes it impossible for the phase to shift its onset in a temporally precise way to the timing onset of the external input resulting in decreased ITPC. Cognitively, this may result in the predominance of internally oriented cognition over externally oriented cognition, with the latter being temporally integrated within the former [38].

Finally, it shall be noted that, while both measures ITPC and ACW are related to input processing, they operate on different temporal timescales. ITPC is calculated in the hundreds-of-millisecond timescale in the period after stimulus onset and hence represents shorter and more fine-grained timescale [12]. In contrast, ACW is calculated either over the entire time series [18] or from several-second segments and therefore represents longer and more coarse-grained timescales [20]. Our ACW–ITPC correlation in healthy subjects suggests a relationship among longer and shorter timescales in the brain’s neural activity: the longer timescales (ACW) of the resting-state shape the shorter ones (ITPC) during task states. That relationship seems to be disrupted in schizophrenia as indicated by the lack of correlation. The deficiency of shorter timescales in the resting-state might thus translate to a deficiency in temporally segmenting incoming information—this is manifest in deficient phase resetting to the stimulus, i.e., decreased ITPC. More generally, this means that the neural activity’s temporal windows can no longer be employed in a temporally precise way, that is, at stimulus onset. However, more research is needed to confirm this mechanism.

In terms of linking symptoms to their underlying neural mechanisms, our corroborating evidence for temporal imprecision could be a first step towards explaining temporal fragmentation experienced by SCZ patients. Often patients will report that their perception of the flow of time is more akin to snapshots rather than a continuum [39,40]. When neural mechanisms that match outer time with inner time are deficient, the world cannot be represented correctly, and patients will experience the discrepancy as sudden jumps in the flow of time. 

### Future Directions

A recent rating scale for the time experience in SCZ provides the possibility to measure temporal fragmentation in psychotic disorders (STEP scale; [40]). It remains to be seen whether temporal imprecision on the neuronal level is related to these specific features of the disorder. Future studies that use the STEP scale and allow for a comparison of resting-state INTs and task-state ITPC are needed to investigate the relevance and potential of the neuronal measures for bridging the gap between brain and experience [40]. It also remains to be investigated whether the dissociation between INTs and ITPC holds for other psychotic disorders, such as bipolar and schizoaffective disorders. Decreased phase clustering in response to stimuli have been reported across the psychotic disorders, albeit to a lesser extent [10,11]. However, to date, no study has used INTs to investigate either bipolar or schizoaffective disorder. Temporal precision can be measured using different methods, such as signal-to-noise ratio and trial-to-trial variability in response to stimuli [3,4]. Future research will have to explore whether a similar relationship between INTs and other task-based measurements exists or if the relationship is specific to INT-ITPC. Testing the relationship we established in this empirical paper with computational modelling, thereby establishing a necessary relationship, could further strengthen the interpretability of our results in the future.

Some limitations should be mentioned. It is unclear how ACW and ITPC are related to each other in healthy subjects. We report, for the first time, their negative relationship in healthy subjects while that is no longer the case in schizophrenia. This leaves open whether the prolongation of ACW in schizophrenia is the cause of the abnormally reduced phase coherence or if both are influenced by similar underlying mechanisms. Additionally, given an unbalanced distribution of the sample in terms of their sex, i.e., an imbalance tilted towards male participants in the schizophrenia groups, findings might not generalize to female schizophrenia patients. Yet, another open question is how this relationship holds at different levels of the sensory processing hierarchy [23]. Our findings pertain exclusively to EEG sensor space and hence make any inference about topography of the relationship impossible. Moreover, it is not clear how the observation of prolonged INTs in EEG [24] can be reconciled with the observation of shortened INTs in fMRI of schizophrenia [23]. Whether that discrepancy is due to the different frequency ranges of EEG (1–80 Hz) and fMRI (0.01–0.1 Hz) remains to be investigated. A final limitation of this study is the use of the FASTER method which interpolates the signal and hence does not result in a genuinely continuous signal [30]. This might have had an influence on calculation of ACW. However, it can be argued that choosing not to interpolate the signal might result in even greater artifacts resulting from muscle activity or blinks. Removing epochs that contain artifacts will result in temporal discontinuities in the signal that would make the acquisition of ACW impossible. 

## 5. Conclusions

We here extend previous findings on abnormal intrinsic neural timescales (INTs) and input processing in schizophrenia. Our study highlights and specifies the assumption of a basic temporal disturbance in schizophrenia on different temporal levels of input processing, namely phase synchronization and intrinsic neural timescales. In schizophrenia, imprecision in response to a stimulus seems to be related to the brain’s capacity to use its own intrinsic neural timescales to segment events in the external world. It remains to be investigated how the shift from short to longer intrinsic neural timescales in the resting state relate to other measures of temporal imprecision and input processing, such as prediction mechanisms and neural variability.

## Figures and Tables

**Figure 1 brainsci-13-00695-f001:**
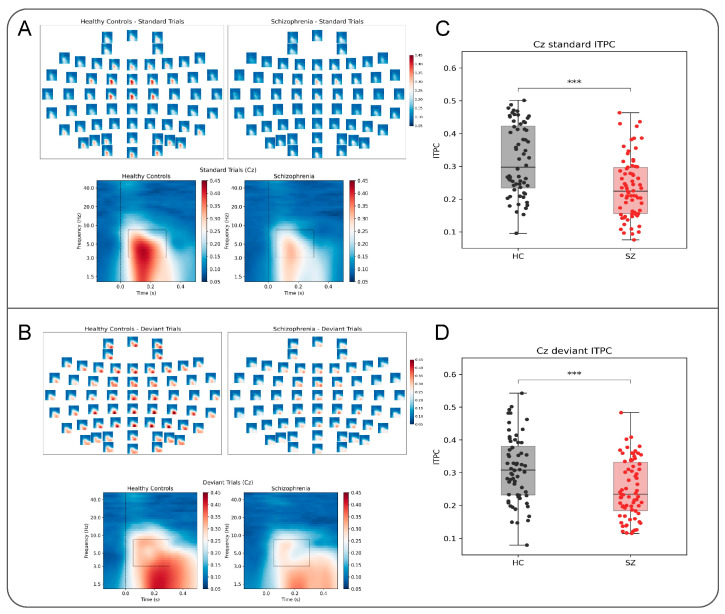
Topography of intertrial phase coherence as well as coherence in electrode Cz for standard (**A**) and deviant (**B**) tones and their group differences (**C**,**D**), respectively. The black boxes in (**A**,**B**) indicate the range for which ITPC was calculated by averaging over values in time and frequency domain; *** < 0.001.

**Figure 2 brainsci-13-00695-f002:**
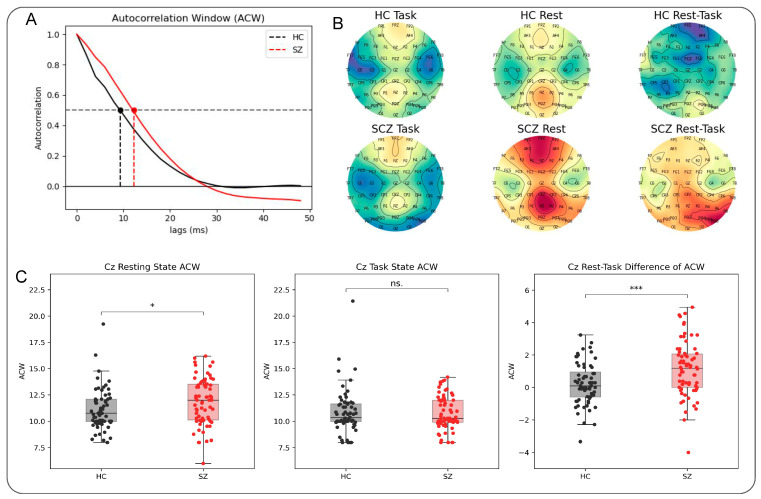
(**A**) Depiction of autocorrelation functions for one healthy participant and one schizophrenia patient. Autocorrelation was measured as the lag where the decay of the autocorrelation function reaches 50 percent. (**B**) Topographical distribution of ACW calculated from resting and task states as well as rest–task difference. (**C**) Group differences in ACW rest, task and rest–task difference; * < 0.05, *** < 0.001, ns. = not significant.

**Figure 3 brainsci-13-00695-f003:**
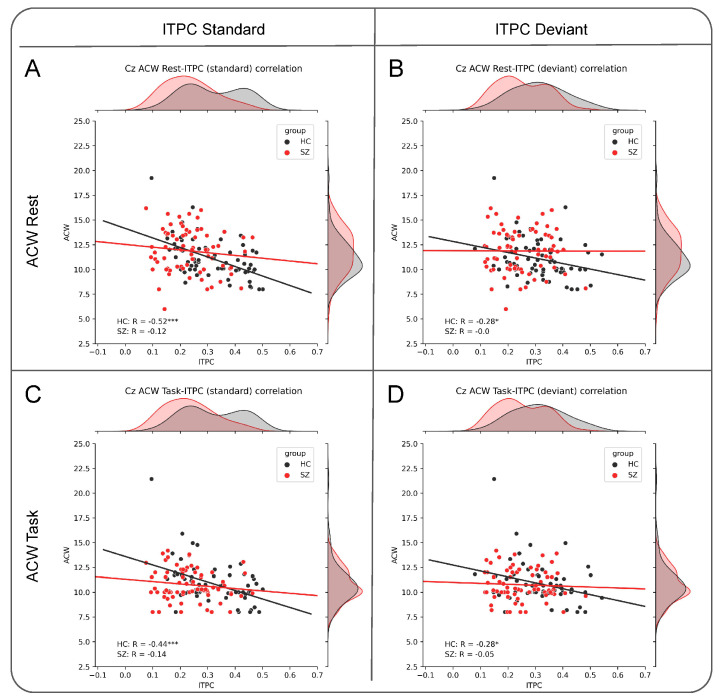
Correlation of ITPC standard (left column, (**A**,**C**)) and deviant (right column, (**B**,**D**)) tones with ACW measured during the resting state (top row, (**A**,**B**)) and during auditory oddball task (bottom row, (**C**,**D**)); * < 0.05; *** < 0.001.

**Table 1 brainsci-13-00695-t001:** Participant demographics, sub-diagnosis and means of all measures of both groups.

	HC	SCZ	
N	64	70	
Sex (F/M)	37/27	22/48	χ^2^ = 9.44, *p* = 0.002
Age	37.62 ± 1.41 years	36.56 ± 1.31 years	*t*_133_ = 0.59, *p* = 0.552
Sub-diagnosis			
Paranoid		39	
Undifferentiated		14	
Residual		4	
No sub-diagnosis		13	
ACW REST	11.02 ± 1.78 ms	11.87 ± 2.27 ms	*t*_133_ = −2.16, *p* = 0.016
ACW TASK	1.78 ± 1.66 ms	1.74 ± 1.57 ms	*t*_133_ = 0.365, *p* = 0.642
ACW REST–TASK DIFFERENCE	0.24 ± 1.27 ms	1.13 ± 1.78 ms	*t*_133_ = −3.38, *p* = 0.001
ITPC STANDARD	0.23 ± 1.27	1.13 ± 1.78	*t*_133_ = 4.8, *p* = 0.0
ITPC DEVIANT	0.32 ± 0.1	0.25 ± 0.08	*t*_133_ = 3.8, *p* = 0.0

## Data Availability

The B-SNIP 1 dataset used in this study is available upon reasonable request at https://nda.nih.gov/edit_collection.html?id=2274 (accessed on 3 December 2020).

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
