# Peer review of "Prolonged Intrinsic Neural Timescales Dissociate from Phase Coherence in Schizophrenia"

_brainsci, 2023, doi:10.3390/brainsci13040695_

Round 1

Reviewer 1 Report

Comments and Suggestions for Authors

This is a very good paper focusing on the investigation of input processing on the brain, mainly on prolonged intrinsic neural timescales dissociate from phase coherence in schizophrenia. The paper is well written and of interest for the readers. However, several minor changes should be made. 

ABSTRACT

1- Before reporting the findings ("our findings show"), the authors should clarify the main aims of the study, and the methods used to reach these findings.

2- At the end of the abstract, the authors report that the findings provide further evidence. This sentence should be rephrased in order to provide future directions based on these findings.

INTRODUCTION

1- Before starting with the topic of "input processing" the authors should introduce several concepts about schizzophrenia: etiological and clinical aspects.

2- EEG should be explained before providing the abbreviation.

3- Lines 74-89: The main aims of the paper should be moved to a specific section, for example: 1.1. Aims. They have been well-described.

METHODS

1- Women were underrepresented in the sample of patients with schizophrenia and were more frequent in the group of healthy controls. This is a limitation of the study that should be described in the limitations section.

RESULTS

1- The findings are well described and presented. Figures are really good.

DISCUSSION

The discussion section needs more references and further discussion. I recomend to add a couple of references for each finding.

A subsection about future perspectives is needed.

CONCLUSIONS

- These should not be a summary from the results, and should be focused on future perspectives of input processing.

Author Response

Thank you very much for the excellent suggestions, we followed up on that and inserted several things into our manuscript. We address each point as it it arouse in your review. I attach a the updated manuscript and will refer to each point by old line-number (OL) and new-line number (NL) for you to conveniently pinpoint the relevant passage in the text. Please also see the comments in the updated manuscript which explicitly stated which item we wished address by our addition. Several small changes were added, mainly punctuation and typesetting. They can be viewed via the reviewing option in Microsoft Word. The updated manuscript entails changes based on comments of both reviewers. Last, due to changes the locations of figues has been rearranged to fit the flow of the text in the updated manuscript.

ABSTRACT

  • Before reporting the findings ("our findings show"), the authors should clarify the main aims of the study, and the methods used to reach these findings.

We added: “The main goal of our paper was to investigate whether reported shifts from shorter to longer timescales are related to decreased ITPC.” (OL: 19 NL: 19)

  • At the end of the abstract, the authors report that the findings provide further evidence. This sentence should be rephrased in order to provide future directions based on these findings.

We added: “which in the future might be able to explain common symptoms related to the temporal experience in schizophrenia, for example temporal fragmentation.” (OL: 28, NL: 31)

INTRODUCTION

  • Before starting with the topic of "input processing" the authors should introduce several concepts about schizzophrenia: etiological and clinical aspects.

AND

  • EEG should be explained before providing the abbreviation.

We added: “Schizophrenia (SCZ) is a psychiatric disorder that is characterized by positive (e.g. hallucinations and delusions) and negative symptoms (e.g. cognitive deficits in attention and working memory). Research on SCZ has been conducted on a plethora of explanatory levels with mechanisms spanning molecular, cellular, circuit and whole-brain system levels. On the system leve , lectroencephalography (EEG) is a non-invasive neuroimaging technique that measuers electrical activity originating in the brain. Obtained signals can be investigated using averaging-techniques such as event-related potentials and time-frequency analysis. Psychiatric research uses EEG to explore electrophysiological mechanisms of disorders and develop biomarkers that can distinguish between them.” at the beginning of the introduction (OL: 33, NL:37)

  • Lines 74-89: The main aims of the paper should be moved to a specific section, for example: 1.1. Aims. They have been well-described.

We added the suggested subsection 1.1 Aims (OL: 74, NL: 90)

METHODS

  • Women were underrepresented in the sample of patients with schizophrenia and were more frequent in the group of healthy controls. This is a limitation of the study that should be described in the limitations section.

The suggested limitations was added in the discussion: “Also, given an unbalanced distribution of the sample in terms of their sex, i.e., an imbalance tilted towards male participants in the schizophrenia groups, findings might not generalize to female schizophrenia patient” (OL: 280, NL: 350)

DISCUSSION

  • The discussion section needs more references and further discussion. I recomend to add a couple of references for each finding.
  • A subsection about future perspectives is needed.

A total of 9 references were added to the discussion and we extended discussion from 62 to 115 lines, including the suggested future perspective subsection. We would ask you to examine the changes in the updated manuscript to see if your comments have been addressed adequately.

CONCLUSIONS

  • These should not be a summary from the results, and should be focused on future perspectives of input processing.

We rewrote the conclusion to be more in line with your comments and would again ask you to examine the changes in the updated manuscript to to see if their comments have been addressed adequately.

Reviewer 2 Report

Comments and Suggestions for Authors

I have read with much interest the article titled " Prolonged intrinsic neural timescales dissociate from phase coherence in schizophrenia ". The author should address some points.

1- Please have a look at the abbreviations all over the article and define them the first time using.

2- The last paragraph in the introduction should be rewritten while shortining it to show only the aims of study

Author Response

Thank you very much for the excellent suggestions, we followed up on that and made changes to our manuscript that hopefully are in line with your intentions. We attach an updated manuscript that contains changes based on both reviews that we received. Below, we also want to address the two points you have raised directly. 

Please also see the comments in the updated manuscript which explicitly stated which item we wished address by our addition. Several small changes were added, mainly punctuation and typesetting. They can be viewed via the reviewing option in Microsoft Word. Last, due to changes the locations of figues has been rearranged to fit the flow of the text in the updated manuscript.

  • Please have a look at the abbreviations all over the article and define them the first time using.

We intoduced the abbreviation  HC in third line of section 2.1. Participants and abbreviation SCZ in a newly added paragraph at the beginning of the introduction. We could not find another missing abbreviation but would be thankful if you could pinpoint any missing abbreviation that we might have missed.

  • The last paragraph in the introduction should be rewritten while shortining it to show only the aims of study

Based on the comments of another reviewer we have separated the aims into a distinct subsection. To address your comment we have removed the sentence “We look at group differences for both standard and deviant tones separately.“ from the section since this does not directly pertain to the aims of the study.